# Rhodamine B-Containing Chitosan-Based Films: Preparation, Luminescent, Antibacterial, and Antioxidant Properties

**DOI:** 10.3390/polym16060755

**Published:** 2024-03-09

**Authors:** Omar M. Khubiev, Anton R. Egorov, Daria I. Semenkova, Darina S. Salokho, Roman A. Golubev, Nkumbu D. Sikaona, Nikolai N. Lobanov, Ilya S. Kritchenkov, Alexander G. Tskhovrebov, Anatoly A. Kirichuk, Victor N. Khrustalev, Andreii S. Kritchenkov

**Affiliations:** 1Department of Human Ecology and Bioelementology, RUDN University, Miklukho-Maklaya St. 6, Moscow 117198, Russia; ihubievomar1@gmail.com (O.M.K.); sab.icex@mail.ru (A.R.E.); darya.semenkova02@mail.ru (D.I.S.); dalialesma01@gmail.com (D.S.S.); asdfdss.asdasf@yandex.ru (R.A.G.); nkumbusikaona534@gmail.com (N.D.S.); lobanov-nn@rudn.ru (N.N.L.); alexander.tskhovrebov@gmail.com (A.G.T.); kirichuk-aa@rudn.ru (A.A.K.); vnkhrustalev@gmail.com (V.N.K.); 2Metal Physics Laboratory, Institute of Technical Acoustics NAS of Belarus, General Lyudnikov Ave. 13, 210009 Vitebsk, Belarus; ilya.kritchenkov@gmail.com; 3Department of General and Inorganic Chemistry, St. Petersburg State University, Universitetskaya Embankment 7–9, St. Petersburg 199034, Russia

**Keywords:** chitosan, rhodamine B, films, photophysical properties, antibacterial activity

## Abstract

In this study, Rhodamine B-containing chitosan-based films were prepared and characterized using their mechanical, photophysical, and antibacterial properties. The films were synthesized using the casting method and their mechanical properties, such as tensile strength and elongation at break, were found to be dependent on the chemical composition and drying process. Infrared spectroscopy and X-ray diffraction analysis were used to examine the chemical structure and degree of structural perfection of the films. The photophysical properties of the films, including absorption spectra, fluorescence detection, emission quantum yields, and lifetimes of excited states, were studied in detail. Rhodamine B-containing films exhibited higher temperature sensitivity and showed potential as fluorescent temperature sensors in the physiological range. The antibacterial activity of the films was tested against Gram-positive bacteria *S. aureus* and Gram-negative bacteria *E. coli*, with Rhodamine B-containing films demonstrating more pronounced antibacterial activity compared to blank films. The findings suggest that the elaborated chitosan-based films, particularly those containing Rhodamine B can be of interest for further research regarding their application in various fields such as clinical practice, the food industry, and agriculture due to their mechanical, photophysical, and antibacterial properties.

## 1. Introduction

Chitosan, a biopolymer derived from chitin, has garnered significant attention in the past few decades due to its attractive characteristics such as biocompatibility, biodegradability, antioxidant effect, and antimicrobial activity [1,2,3,4]. Chitosan-based films have been extensively studied for a wide range of applications, including wound dressings, drug delivery, and food packaging [5,6]. In many instances, properties of chitosan films can be enhanced through the addition of dyes or pigments, to impart new attractive functionalities [7,8]. In the literature, there are numerous examples of chitosan films containing various dyes, such as natural dyes like curcumin and betalains, as well as synthetic dyes like methylene blue and crystal violet [9,10]. The importance and relevance of the development of chitosan-based films has been repeatedly emphasized in recent research papers and reviews [5,11,12,13].

However, among the wide range of chitosan films reported in the literature, there is a growing interest in films containing fluorescent dyes, which have attractive photophysical properties that can open up new possibilities for applications in bioimaging, sensing, and theranostics [14]. While there are several examples of chitosan films incorporated with fluorescent dyes, such as fluorescein and coumarins, there is limited literature on chitosan films containing Rhodamine B, a widely used fluorescent dye [15,16]. Rhodamine B has been extensively studied for its photophysical properties, including its high fluorescence quantum yield and photostability, making it a promising candidate for various applications [17,18]. Therefore, the combination of chitosan films with Rhodamine B holds great potential for developing novel functional materials with enhanced properties (and not necessarily only fluorescent properties, since the new film can demonstrate, for example, improved mechanical parameters, an enhanced antimicrobial effect, and other, sometimes unexpected, characteristics).

In this study, we aimed to investigate chitosan-based films containing Rhodamine B, filling the gap in the literature, and exploring the preparative, photophysical, and biological properties of these films. To the best of our knowledge, there are limited examples of chitosan films incorporating Rhodamine B, highlighting the uniqueness and novelty of our research. We prepared chitosan films containing Rhodamine B using a simple and reproducible method, and characterized the films using various techniques, including mechanical tests, UV/Vis spectroscopy, fluorescence spectroscopy, and X-ray diffraction (XRD). The photophysical properties of the films, such as fluorescence intensity and lifetime, were evaluated. Additionally, we conducted in vitro studies to assess the potential antimicrobial properties of the chitosan films containing Rhodamine B and their antioxidant activity.

The combination of chitosan and Rhodamine B in a film format presents a promising platform for developing multifunctional materials with applications in diverse fields, such as bioimaging, sensing, and antimicrobial and antioxidant coatings (Figure 1). The results of this study, their discussion, and perspectives are presented in the sections that follow below.

## 2. Materials and Methods

Chitosan 40 kDa was purchased from Bioprogress (Russia, Moscow) and Rhodamine B was obtained from Aldrich. Other chemicals and solvents were obtained from commercial sources and were used as received.

Preparation of films: Rhodamine B-containing films and blank films were prepared as follows. An amount of 1.0 g of chitosan was dissolved in 40 mL of 1% aqueous acetic acid, then 0.5 mL of glycerol and 0.5 mL of 0.1% aqueous Rhodamine B were added (no Rhodamine B was added in the case of the blank films). The resultant solutions were cast on Petri dishes and dried at 60 °C (A and A’) and 90 °C (B and B’). C and C’ were prepared as A and A’, followed by 20 min treatment of the film using 20% NH_3_ in EtOH solution and drying at room temperature to constant mass.

Tensile strength and elongation at break were measured using a RAM.1.1.A_RA apparatus (Dongguan, China) at 22 °C (samples 6.0 cm long and 2.0 cm wide).

IR spectra were recorded using a Shimadzu IRSpirit (Kyoto, Japan) at 4700 to 350 cm^−1^ (10 mg of sample without any specified sample preparation).

Differential thermal analysis (DTA) and thermogravimetric analysis (TGA) were recorded using a SDT Q600 (New Castle, USA) using heating rate 5 °C/min in the temperature range from 40 °C to 600 °C.

X-ray diffraction analysis was carried out on a Dron-7 X-ray diffractometer (Saint Petersburg, Russia). A 2θ angle interval from 7° to 40° with scanning step ∆2θ = 0.02° and exposure of 7 s per point were used. Cu Kα radiation (Ni filter) was used, which was subsequently decomposed into Kα1 and Kα2 components during processing of the spectra [19].

Photophysical experiments: UV/Vis light absorption spectra were recorded using a spectrophotometer UV-1800 (Shimadzu, Kyoto, Japan). The emission spectra were registered using an Avantes AvaSpec-2048 × 64 spectrometer (Avantes, Apeldoorn, The Netherlands). The absolute emission quantum yield was determined using an integrating sphere AvaSphere-50 (Avantes, Apeldoorn, The Netherlands). An LED (365 nm) (Ocean Optics, Largo, FL, USA) was applied for pumping. A pulse laser LDH-P-C-405 (wavelength 405 nm, pulse width 50 ps, repetition frequency 10 MHz) (PicoQuant, Berlin, Germany), a photon counting head H10682-01 (Hamamatsu, Hamamatsu, Japan), a multiple-event time digitizer MCS6A1T4 (FAST ComTec, Oberhaching, Germany), and a monochromator Monoscan-2000 (interval of wavelengths 1 nm) (Ocean Optics, Largo, FL, USA) were used for lifetime measurements. Temperature control was performed by using a cuvette sample compartment qpod-2e (Quantum Northwest Inc., Liberty Lake, WA, USA).

Antimicrobial activity (in vitro) was evaluated completely as previously described by some of us [20,21,22]. The DPPH• scavenging effect was evaluated according to the published procedure [23].

## 3. Results and Discussion

### 3.1. Preparation of Rhodamine B-Containing Films

The chitosan-based Rhodamine B-containing films were prepared using the conventional solution casting method. The chitosan which was used in the current study (MW = 40 kDa) is not water-soluble. We dissolved the chitosan in 1% acetic acid solution. Acetic acid protonates primary amino groups of the chitosan, thus, destroying the native interchained hydrogen bonds system. This, in turn, results in the complete dissolution of chitosan.

To improve the flexibility and mechanical properties of the resultant films, we used glycerol as a common plasticizer for polysaccharide-based films. The volume of glycerol added to the chitosan solution was determined based on the literature data to achieve the desired film properties [20]. To the chitosan solution containing the plasticizer, we added a solution of Rhodamine B under vigorous stirring. The initial colorless chitosan/plasticizer solution immediately turned purple. One notable observation was that no heterogeneity was observed in the solution even at a temperature of 80 °C, indicating the uniform distribution of Rhodamine B in the chitosan matrix. This could be attributed to the strong stirring during the mixing process, which ensured a homogeneous dispersion of the dye in the solution. 

In the same manner described above, we prepared the blank films, which were obtained by the same procedures except the addition of the dye Rhodamine B. The resultant solutions were cast in plastic dishes and dried under different conditions which are presented schematically as follows:Dried at 60 °C for 24 h (films A and A’)*;Dried at 60 °C for 24 h and then 90 °C for 2 h (films B and B’);Dried at 60 °C for 24 h, then treatment of the film using 20% NH_3_ in EtOH solution and drying at room temperature (films C and C’);*—the abbreviations A, B, and C belong to the blank films while A’, B’, and C’ belong to the rhodamine-containing films.

Thus, films A and A’ were dried at 60 °C for 24 h, films B and B’ were dried at 60 °C for 24 h, followed by drying at 90 °C for 2 h, and films C and C’ were dried at 60 °C for 24 h, followed by treatment with a 20% solution of ammonia in ethanol (transferring the film to the base form) and air drying to constant weight.

The resultant dye-containing films were transparent purple, while the blank films were practically colorless (Figure 1).

### 3.2. Mechanical Properties of the Films

The most important mechanical parameters for films include tensile strength and elongation at break, which strongly depend on the chemical structure of the components of the elaborated film. Tensile strength is the maximum stress that a material can withstand while being stretched or pulled before breaking. Elongation at break is the ratio of the initial and final lengths of the film before it breaks. Thus, tensile strength is a measure of film strength, while elongation at break characterizes the ductility of the film material. Pure chitosan films are characterized by high strength but very low ductility, so they are dramatically brittle. To avoid this drawback, plasticizers such as glycerin are used. Figure 2 and Figure 3 demonstrate the results of the mechanical tests of the elaborated films.

In many instances, the heating of chitosan films or their transferring into their base form (see Section 3.1, issue 2 and 3) results in changes in the mechanical characteristics of the films, and this is usually explained by different packaging of chitosan macromolecular coils in the films, different water contents, and differences in the hydrogen bonds networks [24]. Thus, heating a blank film for 2 h at 90 °C (A → B) leads to an almost 1.5-fold decrease in strength, and transferring the film to the base form (A → C) results in an approximate 15% increase in strength. In opposition, the heating of the Rhodamine B-containing film (A’ → B’) results in a 2-fold increase in the film strength. Similarly, transferring the Rhodamine B-containing film to its base form (A’ → C’) furnishes a dramatic rise in its strength.

In general, the ductility varies in the opposite direction of strength. Both heating of rhodamine-containing films (A’ → B’) and their transfer to their base form (A’ → C’) reduce ductility by more than five times. Heating the blank films (A → B) strongly increases their ductility. The transfer of blank films to their base form (A → C) decreases their ductility, but only slightly.

Low ductility is an important problem in the mechanics of chitosan-based films. In this regard, we considered film A’ to be the film with the best mechanical characteristics, i.e., the film with the best ductility (the highest elongation at break). The tensile strength of this film is quite sufficient for practical applications in both medicine and the food industry and it is comparable to the strength of similar polyethylene films [25].

### 3.3. Infrared Spectroscopy

IR spectroscopy is based on the absorption of infrared light by a substance, and it is commonly used to estimate the chemical structure of films. In this study, we employed this method to reveal the basic or salt forms of chitosan, as well as to identify a Rhodamine B presence in the elaborated films.

Thus, in all prepared films we were able to identify only the starting chitosan. The spectra of the films, A, B, C, A’, B’, and C’ exhibited stretching vibration bands characteristic for chitosan, i.e., wide bands of O–H and N–H stretching (3440–3100 cm^−1^), C–H stretching (2870 cm^−1^) and bending (1460, 1420, and 1380 cm^−1^) vibrations, and N–H deformation vibrations (1590–1650 cm^−1^). Absorption bands in the range of 900–1200 cm^−1^ are due to C–O–C, C–C, and N–H deformation vibrations. Spectra of samples with the salt form of chitosan (A, A’ B, B’, C’) also show bands characteristic of the protonated NH_3_^+^ group and CH3COO^−^ (1300–1640 cm^−1^) (Figure 4 and Figure 5) [26].

Based on these spectra, it is easy to identify the salt or base form of chitosan by the characteristic structure of the specific vibration bands (1250–1750 cm^−1^), as seen by the moieties highlighted in the corresponding area (Figure 4 and Figure 5). Thus, samples of starting chitosan and C are in base form. Chitosan in the sample C’ was likely partially transferred to its base form. This is apparently because of the significantly lower availability of NH_3_^+^ groups due to the greater compaction of macromolecular coils in the rhodamine-containing film compared to the corresponding blank film. However, the salt form of chitosan in the C’ sample is unambiguously prevailing. Spectra of A’, B’, and C’ do not show Rhodamine B characteristic bands [27] due to its extremely small concentration.

### 3.4. Photophysical Properties of the Films

For all films under study, we investigated in detail the photophysical properties, including the measurement of absorption spectra in the ultraviolet, visible, and near-infrared regions, the detection of fluorescence, the determination of emission quantum yields, and the lifetimes of excited states in different conditions. The results of these measurements are shown below in Figure 6 and Figure 7 and are summarized in Table 1.

The obtained films A’–C’ have an intense color due to the strong absorption of the visible light (more than 90%) with wavelengths less than 600 nm (see Figure 6). Thus, these films transmit only the red part of the visible spectrum (more than 600 nm), which determines their intense red color. This absorption is mostly due to the introduction of Rhodamine B in the composition of these materials (for illustration, Figure 6 shows the spectrum of Rhodamine B in a dilute aqueous solution).

Blank films A–C exhibit strong absorption (over 90% of the light) in the ultraviolet region of the spectrum (less than 380 nm, Figure 6), as well as moderate absorption up to the 500 nm region, which determines that they have only a weak yellow color.

All the films obtained show a noticeable photoluminescence (Figure 7). The quantum yields of this fluorescence are moderate and are approximately 4% for blank A–C films and 2–3% for A’–C’ rhodamine-containing films (Table 1).

Blank films A–C exhibit fluorescence in the blue–green region of the visible spectrum, with band maxima in the region of 490–500 nm. Moreover, the transition from acid films A and B to the basic film C is characterized by an increase in the emission wavelength by 10 nm.

Films A’–C’ containing Rhodamine B show photoluminescence with maxima in the region of 630–640 nm, which is noticeably greater than for Rhodamine B in solution (585 nm, spectra are compared in Figure 7 and Table 1). This phenomenon can be explained by the strong self-absorption of light with wavelengths less than 600 nm in the film with the Rhodamine B molecules [28]. The result of such strong self-absorption is much lower (2–3%) fluorescence quantum yields of loaded films A’–C’ than of free Rhodamine B in solution (31%), despite the fact that the lifetime of the excited state of Rhodamine B in films is longer than in solution (on the order of 4 ns versus 1.6 ns). An additional factor that reduces the quantum yield of the loaded films is the noticeable absorption of light in the ultraviolet region by the film material itself.

The observed hypsochromic shift in the emission maximum of film C’ (633 nm) compared to its analogs A’ and B’ (both 639 nm) is most likely associated with the somewhat lower fluorescence self-quenching in this sample, due to a decrease in absorption with an increase in the pH of the Rhodamine B environment. Nevertheless, this does not lead to an increase in the quantum yield of the film C’, since an increase in pH also leads to a decrease in the luminescence intensity of Rhodamine B [29,30,31,32].

The observed structure of the emission bands of the Rhodamine B-containing films A’–C’ is characterized by the presence, in addition to the main maximum (633–639 nm), of shorter wavelength shoulders in the region of 600 nm, which gives an energy difference between these bands of approximately 1050 cm^−1^, which is in good agreement with the vibration frequencies of aromatic systems in chromophore, and, thus, can be attributed to the vibrational structure of this spectrum. Similarly, a shoulder is observed in the spectrum of free Rhodamine B, but in a longer wavelength region (approximately 630 nm) and with a similar difference in the energies of these bands (approximately 1200 cm^−1^).

Rhodamine B exhibits a significant dependence of the fluorescence intensity and lifetime on temperature changes and is being studied as a temperature sensor [33,34,35,36,37,38,39]. Therefore, it was of great interest to study the dependence of the lifetime of an excited state (a parameter largely independent of concentration, in contrast to intensity) on temperature. As the range of interest, we chose that in the physiological region, i.e., 37 ± 5 °C (see Table 1).

It turned out that blank A–C films show an insignificant response of this parameter to temperature variations (lifetime changes were approximately 0.3–0.9% per 1 °C). In turn, the films loaded with Rhodamine B showed a noticeably higher temperature sensitivity, in the region of 2%. This value is somewhat smaller than for free Rhodamine B in solution (2.4%), which can be explained by the greater rigidity of the chromophore environment in the film matrix.

It should be noted that the decay curves of the fluorescence intensity (which, upon processing, give the values of the excited state lifetimes) in the case of films А’–C’ have the form of a biexponential (or multiexponential) dependence, in contrast to the strictly monoexponential decay of luminescence in the case of free Rhodamine B in solution. Most likely, this fact is explained by the heterogeneity of the state of the Rhodamine B chromophore in the film, its different local surrounding, as well as by possible π–π aggregation, which results in differences in the rates of radiative and nonradiative relaxation [40,41,42].

The emission band maximum in film is longer because of the self-absorption of Rhodamine B (since the Rhodamine B concentration in film is high). As a result, the shorter wavelength part of the emission band of rhodamine in film is partially absorbed by rhodamine itself. The quantum yield of Rhodamine B in film is lower than in solution because of the mentioned above self-absorption of fluorescence shorter than 600 nm by Rhodamine B itself. But the lifetime in this case is independent from absorption (since it is not FRET in its nature). Even vice versa, the lifetime in film is higher because of the rigidity raise due to the insertion of rhodamine in the polymer matrix.

As a result, we demonstrated that the obtained films can be used as luminescent materials and, in the case of films A’–C’ containing Rhodamine B, as effective fluorescent temperature sensors in the physiological range. Therefore, further study and optimization of such composites looks very promising.

### 3.5. X-ray Diffraction Study

The X-ray phase analysis of these samples was carried out on a DRON-7 automatic X-ray diffractometer for polycrystalline materials in the step-by-step scanning mode. A 2θ angle interval from 5° to 45° with scanning step ∆2θ = 0.02° and 5 s exposure per point were used. Cu Kα radiation was used, which was subsequently decomposed into Kα1 and Kα2 components during the processing of the spectra.

Figure 8 and Figure 9 show the diffraction patterns of the studied samples.

Based on the results of the X-ray diffraction studies, we assessed the degree of perfection of the structure of blank and Rhodamine B-containing films. The X-ray diffraction profiles of the amorphous peaks were approximated using the Pseudo-Voigt function. We also refined the peak position, intensity, half-width, and integral peak width. Since the films containing chitosan in its basic form were very hard and had an uneven surface, the peak broadening also occurred due to defocusing of the X-ray beam, so the samples of films in the basic form were excluded from the calculations. As is known, integral broadening is related to the degree of perfection of the structure or broadening due to the size of micro- or nanoblocks. Less integral broadening is evidence of a more perfect structure. Table 2 shows the characteristics of the integral broadening obtained from the results of refinement of the profile of the amorphous peak of the studied samples. For example, in the case of rhodamine films, the integral broadening for films dried at 90 °C (B) was less than for films dried at 60 °C (A). This indicates a more perfect structure of film B and is consistent with its increased strength (see Figure 2 and Figure 3). The same patterns were observed for Rhodamine B-containing films. Moreover, Table 2 clearly shows that the introduction of Rhodamine B into the chitosan-based films increases the perfection of their structure.

### 3.6. Antimicrobial Activity of the Films

Antimicrobial films are of interest in clinical practice and pharmacology [43], in food industry for prolongation of food products shelf-life [12], and in agriculture as plant protecting systems [44]. The elaborated films were tested in vitro as microbial systems toward Gram-positive bacteria *S. aureus* and Gram-negative bacteria *E. coli*, and also toward fungi *A. fumigatus* and *G. candidum*. The results of the biological experiment are presented in Table 3.

Antimicrobial activity of the blank films in salt form A and B exceeds the activity of the blank film in the base form C. This fact can be explained by the enhanced cationic density of chitosan in its salt form. The salt form of chitosan provides its polycationic nature. The chitosan polycation effectively interacts with the anionic regions of the bacterial cell membrane. This interaction results in a cascade of events unfavorable for the bacterium, including disruption of ion pumps, osmotic imbalance, and membrane rupture. All this ultimately leads to the inevitable death of the bacterial cell [45]. Thus, an increase in the cationic density of chitosan entails an increase in its antibacterial effect [26].

Antibacterial activity of the Rhodamine B-containing films, generally, is more pronounced than that of the blank films. This fact is likely due to the presence of Rhodamine B which is characterized by its strong antimicrobial effect [46]. The slightly reduced antibacterial activity of film C can be explained by the fact that this film has partially passed into the base form. The antifungal activity of Rhodamine B-containing films is much more pronounced than their antibacterial activity. The antifungal effect of the blank films is significantly less, therefore, the antifungal effect of films A’–C’ is due to Rhodamine B. The most active antifungal film is A’, which is approximately 50% more effective in comparison with the corresponding blank film. Since it is fungal spoilage that is the main part of microbial spoilage of products, the most antifungal film A’ seems to be the most promising, for example, for protecting food products.

### 3.7. Antioxidant Activity of the Films

On the one hand, one of the reasons for reducing the shelf life of food is oxidative spoilage, so antioxidant active food packaging is of paramount importance in the food industry [47]. On the other hand, pathological processes in wounds and burns are accompanied by oxidative stress. The use of antioxidant systems leads to a decrease in oxidative stress, a decrease in the production of cytokines and inflammatory mediators, and has a beneficial effect on regeneration processes. Therefore, antioxidation films are important in biomedical applications such as wound and burn coatings [48].

In this work, we assessed the antioxidant activity of the prepared films and compared this with a reference highly active antioxidant, i.e., ascorbic acid. The conventional approach to evaluate antioxidant activity is to estimate the capacity to trap reactive DPPH• free radical [49] (Figure 10). The best antioxidant effect is demonstrated by ascorbic acid; at a concentration 1 mg/ml, it scavenges 100% of free reactive radicals DPPH•. The lowest antioxidant effect is characteristic of blank films A–C since they are capable of binding only approximately 40% of DPPH•. The prepared Rhodamine B-containing films A’–C’ display a more pronounced antioxidant activity; at the same concentration, films A’–C’ trap approximately 80% DPPH•. It should be especially noted that within each series A–C and A’–C’, the antioxidant effect does not depend on the method of film processing (drying at 60 °C, or the same followed by 90 °C, or transferring the film to its basic form). In addition, Figure 10 demonstrates that antioxidant activity has a strong concentration dependence; as the amount of any of the tested films in the system decreases, the antioxidant activity decreases. Of course, the increased antioxidant activity of Rhodamine B-containing films is explained by the presence of Rhodamine B in them. The literature data indicate that the capacity to bind reactive DPPH• free radical usually is provided by the H-atom of phenol or aromatic amine functionalities [50,51]. The Rhodamine B molecule contains primary aromatic amino groups, and this explains the increased antioxidant activity of the corresponding films A’–C’.

## 4. Conclusions

In this study, we successfully prepared chitosan-based films and films containing Rhodamine B using the casting method. The films were characterized by their mechanical properties, chemical structures, photophysical properties, and antimicrobial and antioxidant activities. The mechanical properties of the films were significantly influenced by the chemical composition and the drying process. Infrared spectroscopy confirmed the presence of chitosan in all samples, as well as the salt or base form of chitosan.

The photophysical properties of the films were studied, and it was found that Rhodamine B-containing films showed strong absorption in the visible light range, giving them an intense red color, while blank films exhibited strong absorption in the ultraviolet region. All films displayed noticeable photoluminescence. The quantum yields of fluorescence were moderate. Rhodamine B-containing films exhibited a higher temperature sensitivity in comparison to blank films, making them promising candidates for use as fluorescent temperature sensors in the physiological range.

The X-ray diffraction study of the films revealed a correlation between integral broadening and the mechanical properties of the films. The antimicrobial activity of the films was found to be higher for Rhodamine B-containing films, which is likely due to the presence of Rhodamine B and its antibacterial and especially antifungal effect. The films in salt form exhibited higher antimicrobial activity compared to those in their base form. Moreover, Rhodamine B-containing films are characterized by significantly improved antioxidant activity compared with the corresponding blank films.

These results indicate that chitosan-based films, especially those containing Rhodamine B, have promising applications in various fields such as clinical practice, food industry, and agriculture, thanks to their mechanical, photophysical, antibacterial, and antioxidant properties. Further research and optimization of these composites are warranted to enhance their potential uses.

## Figures and Tables

**Figure 1 polymers-16-00755-f001:**
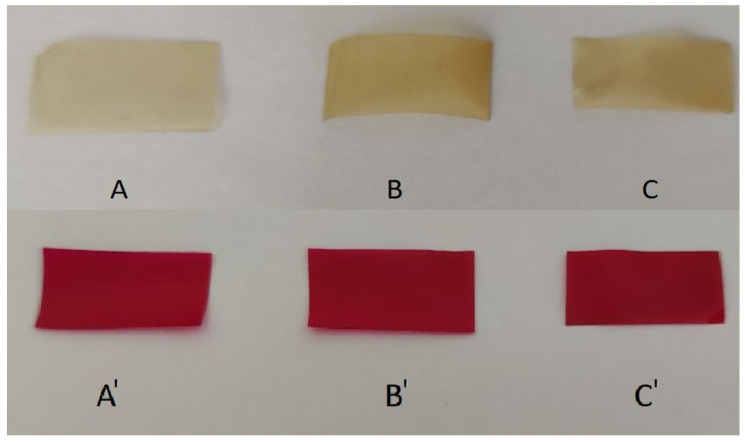
Blank (**A**–**C**) films and Rhodamine B-containing (**A’**–**C’**) films.

**Figure 2 polymers-16-00755-f002:**
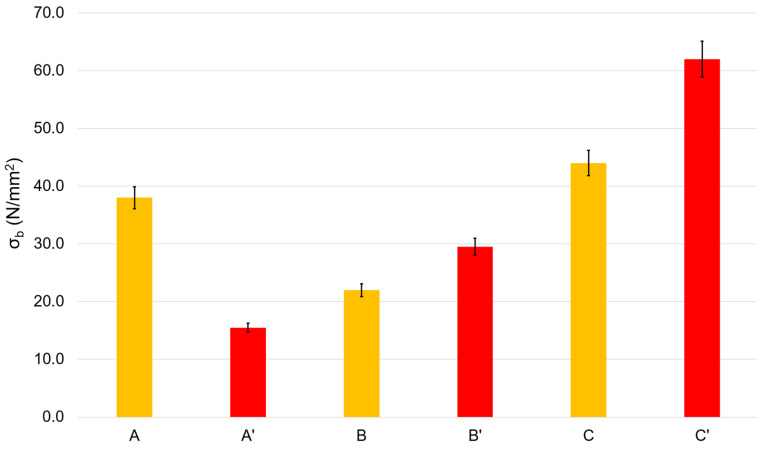
Tensile strength of the blank (A–C) films and Rhodamine B-containing (A’–C’) films.

**Figure 3 polymers-16-00755-f003:**
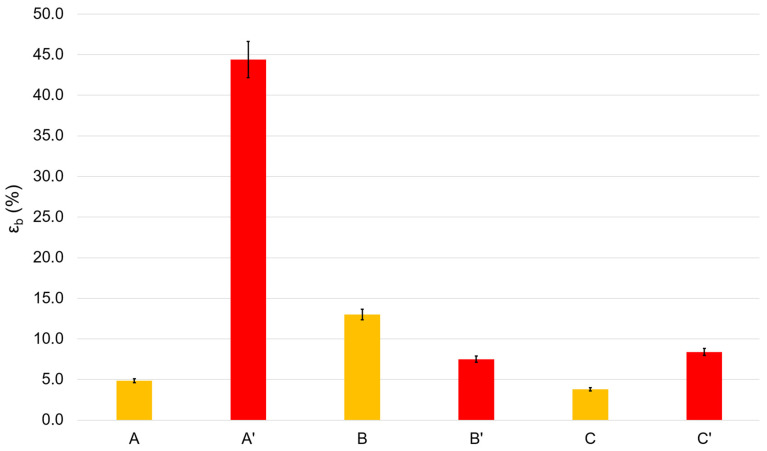
Elongation at break of the blank (A–C) films and Rhodamine B-containing (A’–C’) films.

**Figure 4 polymers-16-00755-f004:**
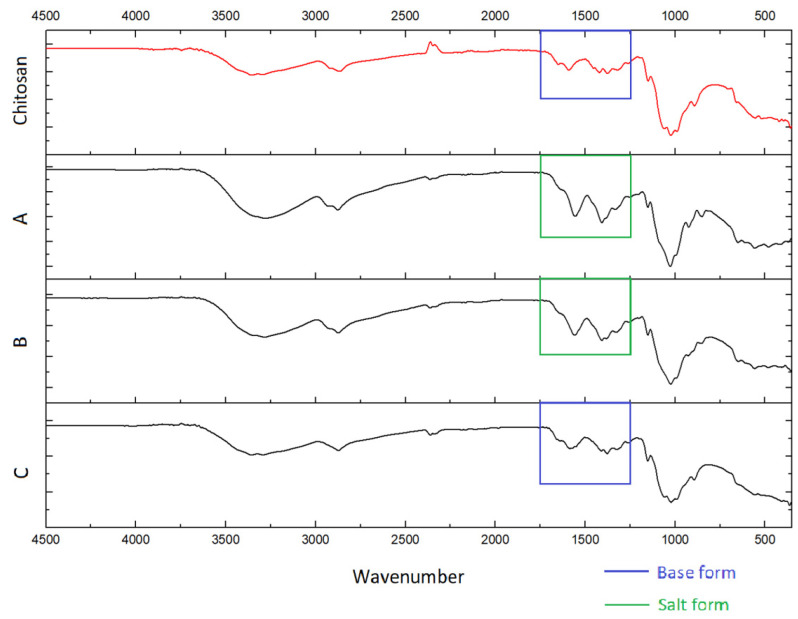
IR spectra of chitosan and blank films A, B, and C.

**Figure 5 polymers-16-00755-f005:**
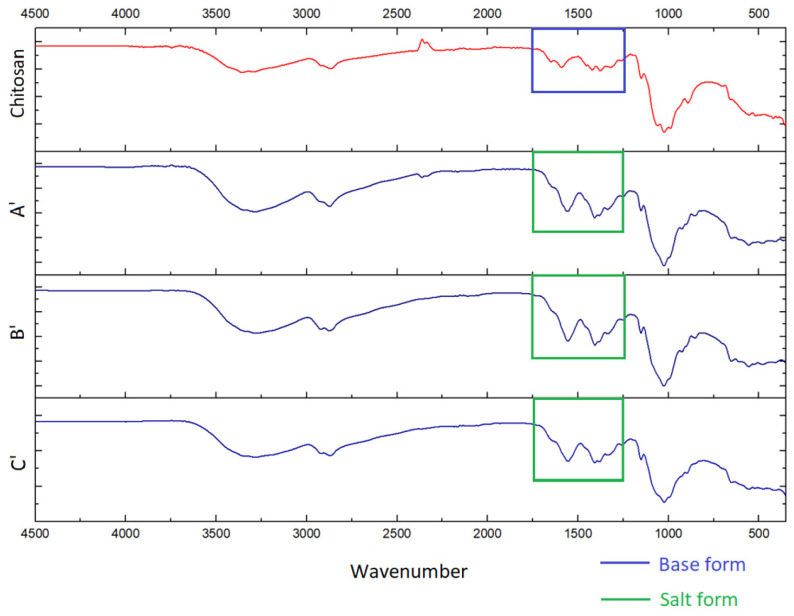
IR spectrum of chitosan and Rhodamine B-containing films A’, B’, and C’.

**Figure 6 polymers-16-00755-f006:**
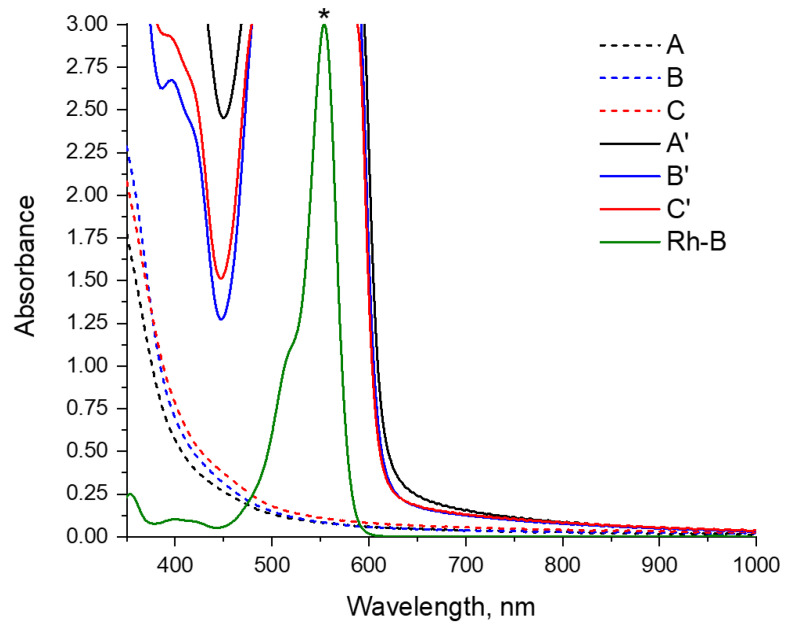
UV/Vis light absorption spectra of films A’–C’, A–C, and Rhodamine B (Rh-B, measured in water solution, C = 1 µM). Temperature 298 K. * Absorbance of Rh-B is normalized to the maximum for the purpose of comparison (absorption data with values greater than 3 are not valid and, therefore, are not displayed).

**Figure 7 polymers-16-00755-f007:**
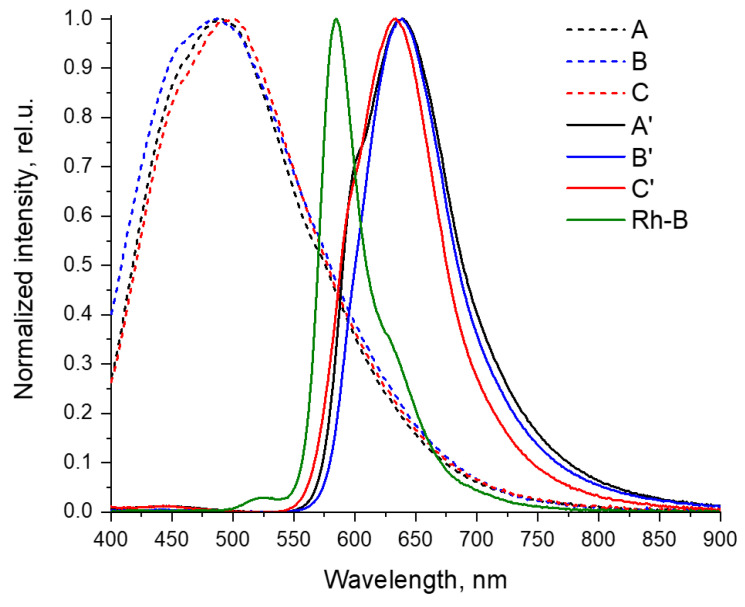
Normalized emission spectra of films A’–C’, A–C, and Rhodamine B (Rh-B, measured in water solution, C = 1 µM). Temperature 298 K. Excitation at 365 nm.

**Figure 8 polymers-16-00755-f008:**
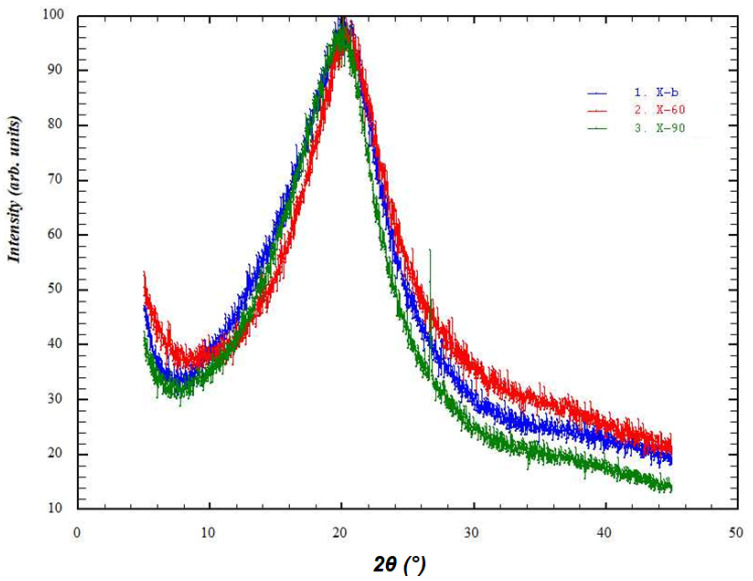
X-ray diffraction patterns of blank films (red—A’, green—B’, blue—C’).

**Figure 9 polymers-16-00755-f009:**
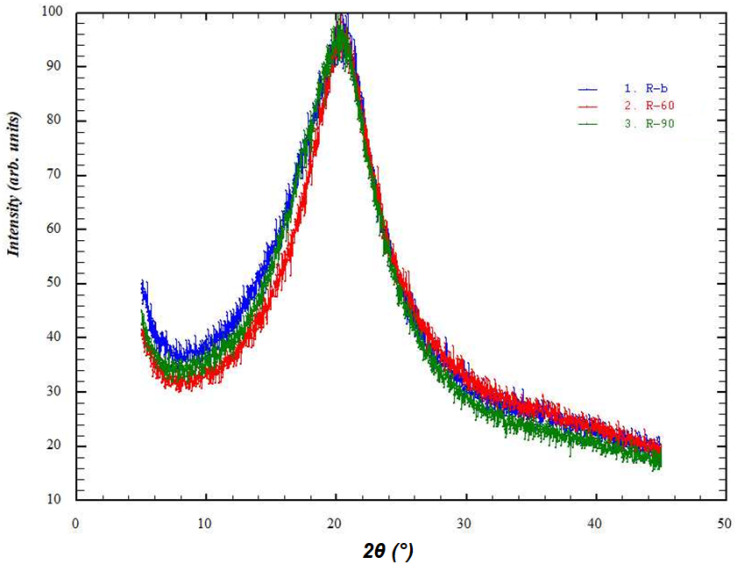
X-ray diffraction patterns of Rhodamine B-containing films (red—A, green—B, blue—C).

**Figure 10 polymers-16-00755-f010:**
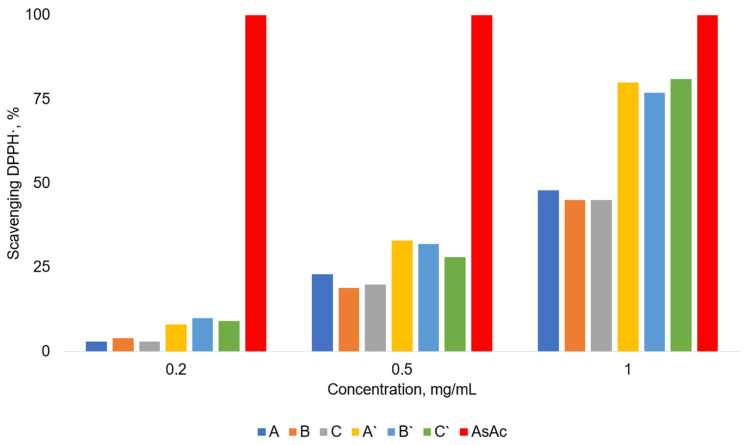
Antioxidant activity of the films (AsAc—ascorbic acid).

**Table 1 polymers-16-00755-t001:** Photophysical data for films A–C, A’–C’, and Rhodamine B (Rh-B, * measured in water solution, C = 1 µM). Temperature 298 K (unless otherwise specified).

	A	B	C	A’	B’	C’	Rh-B *
λem, nm ^a^	452(sh); 490	454(sh); 487	454(sh); 498	601(sh); 639	598(sh); 639	596(sh); 633	585; 630(sh)
QY, % ^a^	4.16	3.91	3.83	1.85	3.34	2.45	31.1
τ, ns (32 °C) ^b^	4.09 ^c^	3.58 ^c^	3.52 ^c^	3.37 ^d^	3.62 ^d^	3.02 ^d^	1.42 ^e^
τ, ns (42 °C) ^b^	3.75 ^c^	3.49 ^c^	3.38 ^c^	2.78 ^d^	3.01 ^d^	2.47 ^d^	1.12 ^e^
∆τ/(τ∙∆T),%/K	0.87	0.26	0.43	1.92	1.82	1.98	2.38

^a^—excitation at 365 nm, ^b^—excitation at 355 nm, ^c^—emission at 495 nm, ^d^—emission at 635 nm, ^e^—emission at 585 nm.

**Table 2 polymers-16-00755-t002:** Integral broadening of the amorphous peak for blank and rhodamine films dried at 60° and heated at 90°.

Sample	Integral Broadening
A	9.76
B	9.51
A’	10.22
B’	10.18

**Table 3 polymers-16-00755-t003:** Antimicrobial effect of the elaborated films.

Sample	Inhibition Zone, mm *			
	*S. aureus*	*E. coli*	*A. fumigatus*	*G. candidum*
A	12.7 ± 0.3	9.2 ± 0.1	11.7 ± 0.2	9.8 ± 0.1
B	12.4 ± 0.2	9.0 ± 0.2	11.8 ± 0.1	9.8 ± 0.3
C	10.3 ± 0.3	7.8 ± 0.2	10.0 ± 0.1	8.6 ± 0.2
A’	14.2 ± 0.1	9.2 ± 0.2	17.4 ± 0.3	16.1 ± 0.1
B’	14.4 ± 0.1	9.2 ± 0.3	16.2 ± 0.2	13.9 ± 0.3
C’	13.7 ± 0.1	8.6 ± 0.2	13.7 ± 0.3	12.7 ± 0.1

* mean value ± S.D.

## Data Availability

Data are contained within the article.

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
