# Peer review of "Rhodamine B-Containing Chitosan-Based Films: Preparation, Luminescent, Antibacterial, and Antioxidant Properties"

_polymers, 2024, doi:10.3390/polym16060755_

Round 1
Reviewer 1 Report
Comments and Suggestions for Authors
Thank you to the author for addressing all the raised comments from reviewers. The authors have carried out the modifications and the corrections to the queries raised and indeed have provided more information in the revised version. Therefore, I recommend this manuscript for publication in Polymers as it is without further modifications.
Comments on the Quality of English LanguageModerate editing of English language required
Author Response
The authors cordially thank the Editor and Reviewers for their unselfish, thorough work, which allowed the authors to amuse the manuscript.
The authors provide answers to reviewers' questions below.
Reviewer 1
Thank you to the author for addressing all the raised comments from reviewers. The authors have carried out the modifications and the corrections to the queries raised and indeed have provided more information in the revised version. Therefore, I recommend this manuscript for publication in Polymers as it is without further modifications.
- Thank you!
Reviewer 2 Report
Comments and Suggestions for Authors
Line 70 – what is it?? Figure 1 is not present here
Line 131 – the word “expect “ is not correct substitute with “except”
Line 147 – the caption is confusing and it is mismacthing what is declared in the text (letters corresponding to films A, A’ etc.)
Lines 162 to 171 – this part is still not clear. Why authors have heated the films and which kind of transformation do they obtain i.e. what does mean transformations like AàB etc.
Comments on the Quality of English Language
I strongly recommend a careful review of English language.
Author Response
The authors cordially thank the Editor and Reviewers for their unselfish, thorough work, which allowed the authors to amuse the manuscript.
The authors provide answers to reviewers' questions below.
Reviewer 2
Line 70 – what is it?? Figure 1 is not present here
- Corrected
Line 131 – the word “expect “ is not correct substitute with “except”
- Corrected
Line 147 – the caption is confusing and it is mismacthing what is declared in the text (letters corresponding to films A, A’ etc.)
- Corrected
Lines 162 to 171 – this part is still not clear. Why authors have heated the films and which kind of transformation do they obtain i.e. what does mean transformations like AàB etc.
- Corrected
Reviewer 3 Report
Comments and Suggestions for Authors
This manuscript has prepared the Rhodamine B reinforced chitosan-based films by the casting method. The mechanical properties, chemical structures, photophysical properties, and antibacterial activities of the film have been characterized. Some major revisions are suggested as below.
(1) It can be seen that films dried at 60℃ and 90 ℃ exhibited different performances, such as mechanical and photophysical properties. Could the author explain why this difference occurs?
(2) Does the content of Rhodamine B have an effect on the properties been studied?
(3) It is recommended to add an error bar to Figure 2 and Figure 3.
(4) The authors mentioned that chitosan in the sample C probably was partially transferred to its base form, please explain why.
(5) Why the maximum emission wavelength of Rhodamine B in solution shorter than the Rhodamine B containing films.
(6) The lifetime of the excited state of Rhodamine B in films is longer than in solution while the fluorescence quantum yields of Rhodamine B loaded films is smaller than free Rhodamine B in solution, why?
Comments on the Quality of English LanguageThe quality of English language is fine.
Author Response
The authors cordially thank the Editor and Reviewers for their unselfish, thorough work, which allowed the authors to amuse the manuscript.
The authors provide answers to reviewers' questions below.
Reviewer 3
This manuscript has prepared the Rhodamine B reinforced chitosan-based films by the casting method. The mechanical properties, chemical structures, photophysical properties, and antibacterial activities of the film have been characterized. Some major revisions are suggested as below.
(1) It can be seen that films dried at 60℃ and 90 ℃ exhibited different performances, such as mechanical and photophysical properties. Could the author explain why this difference occurs?
- Corrected, explanation added in manuscript
(2) Does the content of Rhodamine B have an effect on the properties been studied?
- Corrected, explanation added in manuscript
(3) It is recommended to add an error bar to Figure 2 and Figure 3.
- Corrected
(4) The authors mentioned that chitosan in the sample C probably was partially transferred to its base form, please explain why.
- Corrected, explanation added in manuscript
(5) Why the maximum emission wavelength of Rhodamine B in solution shorter than the Rhodamine B containing films.
- Corrected, explanation added in manuscript
(6) The lifetime of the excited state of Rhodamine B in films is longer than in solution while the fluorescence quantum yields of Rhodamine B loaded films is smaller than free Rhodamine B in solution, why?
- Corrected, explanation added in manuscript
Round 2
Reviewer 3 Report
Comments and Suggestions for Authors
The authors have provided accurate answers and revisions to the questions raised, and I believe that the article has met the requirements for publication.
Comments on the Quality of English LanguageFine